# Understanding Human Preferences:
# Towards More Personalized Video to Text Generation

## ABSTRACT

While previous video to text models have achieved remarkable successes, they mostly focus on how to understand the video contents in a general sense, but fail to capture the human personalized preferences, which is highly demanded for an engaging multimodal chatbots. Different from user modeling in collaborative filtering, there is no other user behaviors in inference as a real-time video stream is coming. In this paper, we formally define the task of personalized video commenting task and design an end-to-end personalized framework for solving this task. In specific, we argue that the personalization for video comment generation can be reflected in two aspects, that is, (1) for the same video, different users may comment on different clips, and (2) for the same clip, different people may also express various opinions with diverse commentary styles. Motivated by these considerations, we design our framework based on two components. The first one is a clip selector, which is responsible for predicting the clips that the user may comment in the video. The second one is a text generator, which aims to produce the comment based on the above predicted clips and the user's preference. In our framework, these two components are optimized in an end-to-end manner to mutually enhance each other, where we design confidence-aware scheduled sampling and iterative inference strategies to solve the problem that the ground truth clips are absent in the inference phase. As the absence of personalized video to text dataset, we collect and release a new dataset for studying this problem. We conduct extensive experiments to demonstrate the effectiveness of our model.

## KEYWORDS

video to text generation, user preference modeling, video comments dataset, personalized content generation, multimodal interaction

## 1 INTRODUCTION

With the advancement of Large Language Models (LLM), LLM based multi-modal models, such as Flamingo [1], BLIP-2 [15], and miniGPT-4 [42], have demonstrated excellent capability to describe an image precisely and with details. Such capabilities are fundamentally useful to test how well a bot can understand vision information; however, they cannot add much value when the bot is expected to accompany human users with shared vision content. Instead of answering some questions on what can be seen, a multimodal bot

*WWW '24, May 13 – 17, 2024, Singapore*
© 2024 Association for Computing Machinery.
ACM ISBN 978-1-4503-XXXX-X/18/06...$15.00
https://doi.org/XXXXXXX.XXXXXXX

is more desired if it can select an appropriate moment based on vision and proactively give an interesting comment, as a family member or a friend can do.

Video to text generation has attracted increasing attention from both academic and industry communities. Standard video to text generation tasks include video captioning [13, 18, 38], video question answering [10, 11, 35], and video comment generation [9, 21, 22, 31, 33]. They mostly focus on how to better represent the videos to generate more accurate texts while neglecting personalization in both selecting an appropriate visual moment and generating a styled text. Recently, some works [24, 28, 37, 41] generate engaging text with diverse style from image. They model user preference by personality traits, user identity, and text style. However, they do not model user preference in video scenarios, without considering user preference combining visual information. Different people may select different parts of a video for captioning and describe them in different ways, which have not been addressed yet.

Taking real-life communication among humans as an example, the human personalities play important roles in their interested clip and generated text. As shown in Fig. 1, different users may comment on various clips of the same video, *e.g.*, $u1$ comments on clip (a) and (b), while u2 comments on clip (a) and (c). Even for the same clip, different users may also have various styled comments. For example, $u1$ usually expresses her opinions by a long text with more details, while $u2$ prefers shorter and more succinct comments. Thus, $u1$ comments clip (a) and (b) by "After a violent reaction, the production is complete" and "The Jerry mouse evolved successfully", respectively, while, for clips (a) and (c), $u2$ only says "the king of dark cuisine" and "So cute". All the above examples suggest that the comments with obvious personal style makes communication more interesting. Personalized style is reflected not only in the textual style but also in the selection of video clips.

To enhance engaging multimodal chatbot performance, in this paper, we formally define the problem of personalized video commenting. While the task seems to be interesting, it is non-trivial due to the following challenges: to begin with, in real scenarios with multimodal bot companionship, visual information is input in the form of a stream rather than ready-made videos. This requires solutions solely make use of users' history and the vision content before this moment. Collaborative filtering methods are not applicable here because there is no comment from other users available in real-time applications. Second, there is few previous work on incorporating user preferences into the video to text generation process. Thus, incorporating what types of user information as user preference and how to incorporate such preference may challenge our model designs. Third, generating comments for a video includes two steps: (1) selecting the clips to comment and (2) generating comments based on the selected clips. Ideally, these two steps should be able to enhance each other. More accurately selected clips can lead to more consistent comments with the ground truth. If one can accurately predict the comments for a clip, then given

---

**Title: Dubbing of Tom and Jerry**

| 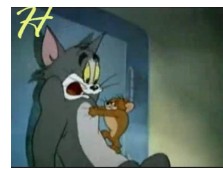 | 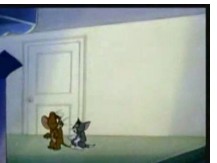 | |
|:---:|:---:|:---:|
| (a) 02:09 | (b) 04:20 | (c) 06:21 |

| | | |
|---|---|---|
| **(u1)** After a violent reaction, the production is complete!
**(u2)** The king of Dark Cuisine.
**(u3)** Perfect proportion!
... | **(u1)** The Jerry mouse evolved successfully.
**(u4)** Make miracles happen with great effort.
**(u5)** Mighty Mouse.
... | **(u2)** So cute!
**(u6)** The efficacy of the potion is not very stable.
**(u7)** Really stupid!
... |

**Figure 1: An example of comments from different users posted on clips of the video – " Dubbing of Tom and Jerry".**

the ground truth, she can also identify which clips the comments should belong to. However, the clip selection process is a discrete operation, how to soften such operation to build an end-to-end framework to mutually enhance the above two steps needs our careful designs.

To overcome the above challenges, we design an end-to-end framework to incorporate the user personalized preferences into the video comments generation process (called **PVCG** for short). In specific, our framework is composed of two components. The first one is a clip selector, which aims to predict the clips that the user would like to comment on by three user preference modeling methods. We input the comment and all the clips of a video, and output the probabilities of all the comment-clip pairs, which are compared with the ground truth clip in the dataset for optimization. The second one is a comment generator, which is responsible for producing the comments according to the above predicted clips and the user preference. To bridge these components, we soften the clip selector by leveraging softmax to weighted average the visual features of all the clips in a video, and the obtained results are input into the comment generator. Since softmax is a differentiable operation, the above two components can be optimized in an end-to-end manner. Considering that the ground truth for the selected clips and user comments are not available in the testing phase, we design the strategies of **scheduled sampling** and **iterative inference** to bridge the gaps between the model training and testing processes.

The main contributions of this paper are as follows:

- We highlight the importance of personalization in the problem of video to text generation, and formally define the task of personalized video commenting in real-time accompanied scenarios.
- To solve the above problem, we propose a personalized framework, which contains a clip selector and a comment generator, and design strategies to enhance the quality of generated text.
- We release a new dataset for studying the problem of personalized video comments generation. We conduct extensive experiments on proposed datasets to demonstrate the effectiveness of our model.

## 2 RELATED WORK

We divide related works into three categories.

### 2.1 Video to Text Generation

Many works focus on exploring video to text generation tasks, including video captioning, video question answering, and video comments. Video captioning [13, 18, 38] takes video frames as input and generates a relevant description. Video question answering [10, 11, 35] asks machines to answer the question based on the video content. Video commenting generates human-like comments according to the video context. LiveBot [22] first proposes the video comment generation task. It constructs a large-scale video comment dataset and generate comments using surrounding comments from other users and visual content. Following LiveBot, Lv et al. [21] proposes an Embedding-based Generative Adversarial framework to bridge the gap between visual and textual content. Also, some works [9, 31] introduce additional audio information, using a multi-modal and multi-task architecture to capture the relationships among comments, vision, and audio, generating comments with these information. Although Wu et al. [33] explores comments generation with few surrounding comments, it still ignores personal preference when selecting video clips and generating comments.

### 2.2 Personalization in Vision and Language Research

Despite many works focusing on standard video to text generation, some works explore personalization in vision and language research to meet users' real demand. AttendToYou [24] incorporate each user's active vocabularies into memory networks to capture their writing style. EICP [28] specifies 215 different personality traits as user preference. MHTH [37] learns user's short-term and long-term literal-preference by user identity and user recent comments respectively. SACO [41] proposes a style-aware triplet contrast learning method to generate text with desired style from image. These works consider the importance of user literal-preference in generating personalized text. However, they ignore the user preference in visual information selection. Some works [3, 32, 34] encode user preference based on both visual and textual information for recommending personalized key frames. But they do not further

explore personalized text generation in video scenarios. Different from an image, a video always contains more diverse information. Thus, different people may select different parts of a video for captioning and describe them in different ways, which have not been addressed yet.

### 2.3 Explainable Recommendation Considering User Preference

Explainable recommendation refers to personalized recommendation algorithms that not only provide users with recommendation results, but also explain why such items are recommended. Similar with personalized video to text generation, it learns to perform rating prediction and user-dependent sentences related to this prediction. Explainable recommendations naturally involve humans in the loop. Similar to personalized video to text generation, explainable recommendation generates user-dependent explanation considering modeling users' behavior.

Based on the different forms of explanation, explainable recommendation systems can be categorized into many types. User/item-based explanations are usually provided based on users' feedback [4, 36]. Feature-based explanation using content-based recommendation methods to provide explanation [12, 39]. Besides, as users' reviews and social media posts contains their opinion, some works show that such information is quite beneficial in user preference modeling and recommendation [19, 27, 30]. Based on the development of natural language generation techniques and computer vision systems, many explainable recommendation systems generate explanation sentences using knowledge from other modality. They utilize item images or social information to generate explanation sentences [2, 20, 25]. User preference modeling is well-discussed in explainable recommendation systems, which can be applied to personalized video to text generation tasks. The difference is that for a new video in our scenarios, there is no other user behaviors available associated with the video, which is more challenging.

## 3 PERSONALIZED VIDEO COMMENTING TASK

In this section, we formally define the new problem of personalized video commenting and describe how we construct a dataset to support such kind of research.

### 3.1 Problem Formulation

We imagine that in the future a chatbot can accompany a lonely user watching films or walking together with camera opened. The chatbot can make funny comments whenever it finds some interesting points. Thus the user will never feel lonely. Moreover, in such a scenario, a chatbot can provide more emotion values if it can play as a character with a personalized taste of selecting an interesting clip and making styled comments.

Suppose we have a user set $\mathcal{U}$ and a video set $\mathcal{V}$. For each video $v \in \mathcal{V}$, we segment it into $K$ clips, that is, $v = \{v_1, v_2, ..., v_K\}$. The user comments are collected in $D = \{(u_i, v_{ik}, t_i)\}_i^N$, where each element $(u_i, v_{ik}, t_i)$ means that user $u_i$ comments on the $k$-th clip $v_k$ of video $v$ with text $t_i$.

Given $D$, our task is to learn a model $f$, such that, for a new video $v^*$, it can accurately predict which clip of the video will be

**Table 1: Overall statistics of the Personalized VideoIC dataset.**

| | |
|---|---|
| Total number of users | 7,065 |
| Total number of videos | 4,036 |
| Total number of clips | 94,831 |
| Total number of comments | 263,835 |
| Ave. comments per video | 65.4 |
| Ave. clips per video | 23.5 |
| Ave. comments per user | 37.3 |

chosen and how it will be commented by each user corresponding to their personality. In order to accomplish this task, the model needs to firstly predict which clip the user may comment on, and then generate the comment contents.

It should be noted that for the new video there is no other users' behaviors available for collaborative filtering. Only the video content and the user's history in training data can be exploited. This is a big difference from user behavior based recommendation.

### 3.2 Personalized Video Commenting Dataset

We build a new dataset to support research on our proposed new task. In specific, our dataset is built upon VideoIC [31], which is a time-synchronized comment (TSC) dataset, containing 4,951 videos and 5 million comments. However, the user IDs are not provided in VideoIC. Therefore, we crawl user IDs and corresponding comments from the website [1] where VideoIC is collected from and construct Personalized VideoIC dataset.

For each video, we segment it into clips every ten seconds, and each record in our dataset is a tuple $(u, v, c, t)$, where $u$ is a user, $v = \{v_1, ...v_K\}$ is a set of clips in a video, $c$ is the index of the user selected clip and $t$ is the comment posted by the user. If a user has multiple comments for a video, then we separate them into different records. Thus only one clip is commented in each record. We remove users with less than twenty comments and their corresponding records.

The final statistics of our dataset are summarized in Tab. 1. We split the data into 3636, 200, and 200 videos for training, validation, and testing respectively. The videos in the test set never appear in the training set or validation set, which ensure they are new. Unlike certain personalized image to text datasets [28], our Personalized VideoIC dataset does not rely on personality traits that are manually labeled by humans. A user's personality is implicitly represented by her selected clips and posted comments in the training set. If a method can better model the user, it will achieve better performance in predicting her selected clips and generated comments in the test set.

## 4 OUR PROPOSED PVCG MODEL

In this section, we introduce our model for solving the personalized video commenting problem in detail.

---

[1]https://www.bilibili.com/

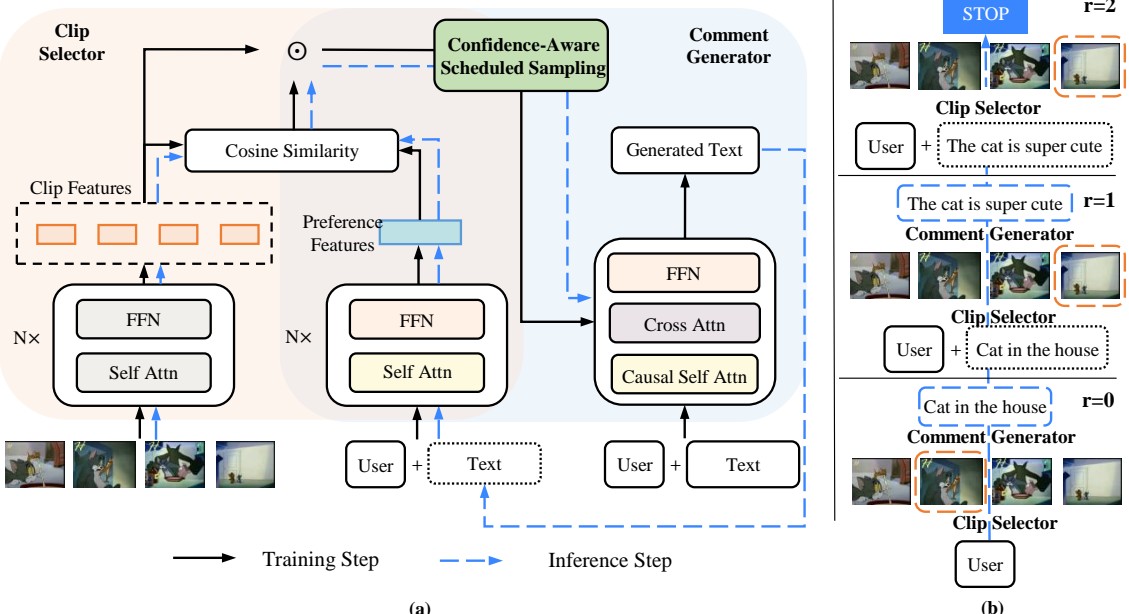

**Figure 2: Overview of our proposed PVCG framework. (a) shows that the PVCG contains two modules: clip selector and comment generator. The black solid line represents the training stage, and the blue dashed line represents the inference stage. (b) illustrates iterative comment inference through an example. It starts with a user preference feature to select a clip, here is the second one, and then generates a comment "cat in the house" in the first round. Then concatenating the user ID with the generated comment can select the fourth clip and then generate another comment "the cat is super cute" at the second round. Finally concatenating the user with the new comment still selects the fourth clip at the top and thus stops the process of iterative comment inference at the third round.**

## 4.1 Framework Overview

As shown in Fig. 2 (a), our proposed PVCG framework consists of two main components: the clip selector to predict the clip that the user may comment in a given video, and the comment generator to produce the comments on the predicted clip. The clip selector can match clip features with user preference features by optimizing a contrastive loss during pre-training and further select users' most interested clip in a given video by optimizing a cross-entropy loss. The comment generator is designed to estimate textual comment of a user on the selected video clip and user preference by optimizing a generation loss. In the example shown in Fig. 2 (b), the clip selector may select the second clip and the comment generator generate "Cat in the house" in the first round.

We argue that these two components can mutually enhance each other to predict the final user comments. On one hand, based on an accurate enough clip selector, the comment generator can find more accurate visual features of the clips, and thus can predict better comments. On the other hand, if the comment generator is sufficiently well, then the clip selector can be more accurately supervised, since a wrong clip can be more likely to lead to an inaccurate comment.

If the clip selector and comment generator are optimized separately, it fails to influence and mutually enhance each other. To build a bridge between them, we soften the prediction results of the clip selector, and leverage them as the input of the comment generator to derive an end-to-end framework and propose a confidence-aware strategy to schedule the process of training the model. In addition,

we propose an iterative strategy to inference comment. Thus the generated comment in the first round can provide more information to user preference features and thus select a better matched clip in the second round. Later the clip can generate better comment then. For example, in Fig. 2 (b), the fourth clip, which is the ground truth clip, is selected in the second round. Thus the comment generator outputs a new comment "The Chat is super cute". Finally, the iterative comment inference stops in the third round when the selected clip is the same to that selected in the second round.

## 4.2 User Preference Modeling

User preference for a given video is related to both user's static interest and the content of current video clips. Thus, combining user's static preference and their dynamic preference for current video is crucial in modeling user preferences. Based on this, there are three methods to model user preferences.

(1) **User identity.** Users who post their comments online are typically associated with user IDs. We aim to leverage the IDs to learn static user preference. As shown in Fig 3 (a), we formulate the input format as "The comment of user <user id>" to indicate the user identity in both training and inference stage.

(2) **User's historical behavior.** Another way to model user's static preference is leveraging their historical behavior/interests. Here, we consider the video clips that users have commented on, along with their corresponding comments, as their historical interests. As shown in Fig 3 (b), we use an image encoder and a text

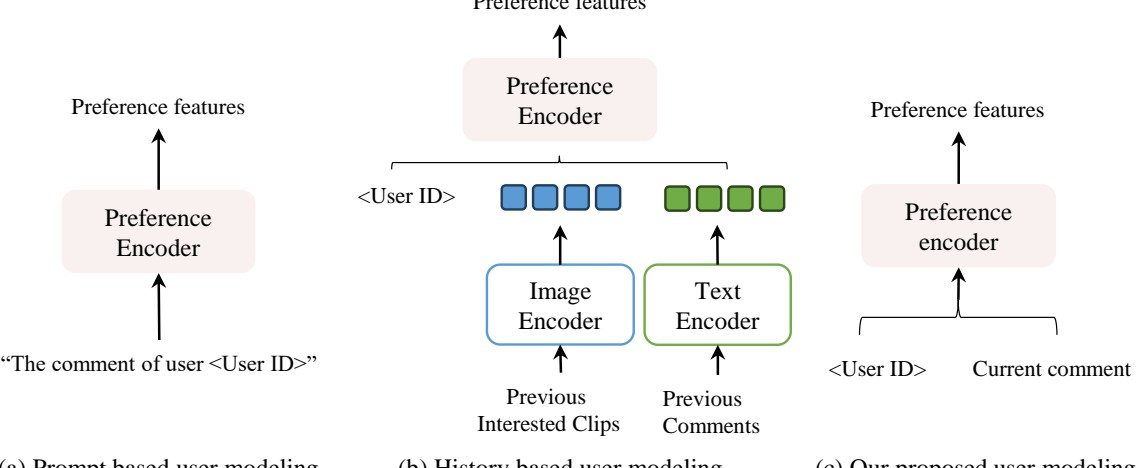

Figure 3: Illustration of different user preference modeling method.

encoder to encode these historical information respectively and concatenate them into a sequence to model users' preference.

(3) **User identity with current comment.** User's preference is not only related to her/his static preference, but also related to current videos. We map user identity to learnable user embedding, learning user's static preference. Also, we model user's dynamic preference from user's comments for their interested clips (As shown in Fig 3 (c)).

## 4.3 Clip Selector

The clip selector aims to predict where the user may comment on a video. The inputs include two parts: (1) the visual embeddings of all the clips of a video, and (2) the user preference features. The output is the distribution that reflects the selection preference of the user on the clips. Formally, for a video $v$ with clips $\{v_1, v_2, ..., v_K\}$, we firstly use a visual transformer [8] to project each clip $v_k$ into an embedding, that is:

$$e_{v_k} = \text{ENC}_V(v_k), \tag{1}$$

where $\text{ENC}_V$ is implemented based on the architecture introduced in [8]. $e_{v_k} \in \mathbb{R}^d$ is the obtained $d$-dimensional visual embedding for $v_k$. We use $E_v \in \mathbb{R}^{K \times d}$ to represent the embeddings of all the visual features of the clips in $v$.

For a user $u$, suppose she has posted a comment $t$ for the $k$-th clip of video $v$, then we leverage the preference encoder to derive the user preference features. It learns both dynamic user preference which is related to current video and static user preference which is consistent among all videos. Here we leverage user ID to learn static preference and model dynamic user preference from comment $t$. The preference encoder is implemented based on a deep bidirectional Transformer [7], where we use a learnable look-up table mapping user ID to embedding, and append it before the comment $t$'s word embedding sequence, that is:

$$e_u = \text{ENC}_U(u, t), \tag{2}$$

where $\text{ENC}_U$ is the encoder for generating the user preference features $e_u$.

The training of clip selector can be divided into two stages: In the first stage, we want to make sure the clip selector have the ability of understanding visual information and user preference representation. Therefore we use contrastive learning by encouraging positive clip-preference pairs to have similar representations in contrast to the negative pairs. We regard user preference features and clip features which the user is interested in as the positive pair, while the others in a mini batch are negative pairs. We sample negative clips across the entire dataset, rather than just sampling within the same video to avoid false negatives. In the second stage, we encourage clip selector to learn user's most interested clip in a given video. For the dataset $D$, we optimize the clip selector based on the following cross-entropy loss:

$$\mathcal{L}_{select} = - \sum_{(u, v_k, t) \in D} z_k \log g(e_{v_1}^T e_u, e_{v_2}^T e_u, ..., e_{v_K}^T e_u), \tag{3}$$

where $z_k$ is a K-dimensional one-hot vector with the $k$-th element equal to 1. $g$ is the softmax operator.

## 4.4 Comment Generator

The comment generator $\text{DEC}_G$ aims to estimate the textual contents of the comment of a user on a video clip. Formally, suppose the visual embedding of the clip is $\hat{e}$, then the comment is generated based on a multi-modal text decoder with the user embedding as the initial input. We insert a cross-attention layer between the casual self-attention layer and the feed forward layer to integrate user and clip information $\hat{e}, e_u$ into the generated comment.

Suppose the words in comment $t$ is $\{t_1, t_2, ..., t_l\}$, then the objective for optimizing the comment generator is:

$$\mathcal{L}_{gen} = - \sum_{(u, v_k, t) \in D} \sum_{i=1}^{l} \log p(t_i | t_{<i}, \hat{e}, e_u), \tag{4}$$

where $p$ represents the probability of the next word given the previous ones based on BERT.

Instead of optimizing the two losses separately, we soften the prediction results of the clip selector, and concatenate user preference features as the input of the comment generator to derive an

end-to-end framework. Formally, we let

$$\hat{e} = \sum_{k=1}^{K} [g(e_{v_1}^T e_u, e_{v_2}^T e_u, ..., e_{v_K}^T e_u)]_k e_{v_k}, \tag{5}$$

where $[g(e_{v_1}^T e_u, e_{v_2}^T e_u, ..., e_{v_K}^T e_u)]_k$ is the $k$-th element of the softmax output. This equation actually weighted averages all the embeddings of the video clips.

Since softmax is a differentiable operator, the final supervision signal (*i.e.*, the user comments) can be back propagated to influence the parameters of the clip selector via $\hat{e}$. Finally, the complete objective of our model is:

$$\mathcal{L} = \mathcal{L}_{select} + \mathcal{L}_{gen}. \tag{6}$$

## 4.5 Confidence-Aware Scheduled Sampling

While in the above formulation, the clip selector and comment generator can be optimized in an end-to-end manner, the weights $[g(e_{v_1}^T e_u, ..., e_{v_K}^T e_u)]_k$ leveraged to derive $\hat{e}$ can be inaccurate in the initial stage of model optimization, which may lower the final performance. To solve this problem, we propose a confidence-aware strategy to reschedule the model training process. Our general idea is that, when the confidence of the clip selector is not sufficiently high, we use the ground truth signal to guide the comment generator learning process. When the clip selector is more confident for its predicted results, then we use $\hat{e}$ as the input of the comment generator, which is more consistent with the model inference stage. Formally, we use the loss $L_{select}$ to evaluate the confidence of the clip selector. Then, we have:

$$\hat{e} = \begin{cases} \sum_{k=1}^{K} z_k e_{v_k}, & \text{if } L_{select} > \theta, \\ \sum_{k=1}^{K} [g(e_{v_1}^T e_u, e_{v_2}^T e_u, ..., e_{v_K}^T e_u)]_k e_{v_k}, & \text{else,} \end{cases} \tag{7}$$

where $\theta$ is a threshold to measure whether the confidence of the clip selector is high enough.

The above strategy simultaneously considers the prediction error of the clip selector as well as the consistency between the model training and testing phases. If we always use the ground truth clip as the input of the comment generator, the learned model can be incompatible with the testing environment, where we do not know the clip ground truth. If we leverage $\hat{e}$ as the input of the comment generator in the complete optimization process, then for the cases where the clip selector is less confident, the comment generator may receive inaccurate input signals, which may impact the model optimization. Our designed strategy makes a trade-off between the above two settings, which is shown to be effective in the experiments.

## 4.6 Iterative Comment Inference

In Eq. 2, the preference feature is derived based on the ground truth comment, which is only available in the training phase. To make the inference, in the first round we only input $u$ into $\text{ENC}_U$, and left $t$ as an empty set. Then, we use the obtained embedding $e_u$ to generate a comment $t_0$. In the next round, we input $u$ and $t_0$ into $\text{ENC}_U$ to generate a new comment $t_1$, which is then input

---

**Algorithm 1** Iterative Comment Generation

**Data** : user $u_i$, a new video $v^* = \{v_1^*, ..., v_K^*\}$, and interation round $r$.
**Result** : clip $v_p^*$ user may comment on, and corresponding text $t_i$.

1: $r \leftarrow 0$
2: $t_i \leftarrow None$
3: **while** not convergence **do**
4:    $r \leftarrow r + 1$
5:    $e_{v_k} = \text{ENC}_V(v_k)$
6:    $e_u = \text{ENC}_U(u, t_{i,p})$
7:    $\hat{e} = \sum_{k=1}^{K} [g(e_{v_1}^T e_u, e_{v_2}^T e_u, ..., e_{v_K}^T e_u)]_k e_{v_k}$
8:    $\hat{t}_i = \text{DEC}_G(\hat{e}, e_u)$
9:    $t_i \leftarrow \hat{t}_i$
10: **end while**

---

into $\text{ENC}_U$ again to generate $t_2$. After $R$ round iteration, we use the output $t_R$ as the final predicted user comment. The complete iterative comment inference process can be seen in Algorithm 1.

The strategy of iterative comment inference bridges the gap of the inconsistent inputs of $\text{ENC}_U$ in the training and testing stages. Unlike previous methods [14, 28, 41], which only use an empty set to replace $t$, we further leverage the generated comments $t_1$, $t_2$, ... to enhance the inference stage. Ideally, as the number of iterations becomes larger, the inference stage is more consistent with the training phase, which facilitates more accurate user comment generation.

## 5 EXPERIMENTS

We focus on the following research questions:

**RQ1**: Whether our framework can achieve better performance for the task of clip selection?

**RQ2**: Given the user actually commented clips, whether our framework can achieve satisfactory performance for the task of comment generation?

**RQ3**: Without the ground truth clips, whether our method can achieve better performance in comment generation task?

## 5.1 Experiment Setup

*5.1.1 Evaluation of Clip Selection.* To evaluate how well a method can select a user's interested clip, we form it as a ranking problem and use Recall@$k$ to measure. Recall@$k$ means how many percentage of records a method can retrieve the ground truth clip at top $k$. We implement baselines with different user preference modeling:

- **MostPopular** This is a non-personalized method. We fine-tune BLIP [16] on the Personalized VideoIC dataset, without using user identity information. In the inference stage, it generates comments using most popular clip of the video. Although in our proposed task, there are no other users' behavior available for a new video, we still implement this method by leaking other users behavior as a reference.
- **BPR-MF** This is a personalized recommendation based method with implicit feedback. Here, we treat clips that users commented as positive items and those they didn't commented as negative items. The clips are ranked based on users' history behaviors,

**Table 2: Comparison of clip selection accuracy. † and ∗ denote significant improvements over the Pers (history), PVCG (cascaded) methods results with $p$-value< 0.05 in t-test respectively. Here, "Pers" refers to "personalized."**

| Settings | Recall@1 ↑ | Recall@5 ↑ | Recall@10 ↑ |
|---|---|---|---|
| MostPopular | 3.45 | 10.17 | 22.99 |
| BPR-MF | 6.18 | 13.96 | 36.04 |
| Pers (user id) | 8.04 | 26.44 | 52.87 |
| Pers (history) | 18.06 | **44.44** | 59.72 |
| PVCG (cascaded) | 18.39 | 32.18 | 60.92 |
| PVCG (r=0) | 25.35 | 40.85 | 60.56 |
| PVCG (r=4) | **27.54**$^{†*}$ | 42.18$^{*}$ | **65.92**$^{†*}$ |

without considering any comments information. The code is implemented through RecBole [40].

- **Personalized (user id).** It adopts a prompt-like mechanism to incorporate user preference information (as shown in Fig. 3 (a)). The structure of the user preference encoder is the same as PVCG, but uses prompts instead of user identity as the input. The prompt is set as "The comment of user ⟨user id⟩" in both training stage and inference stage.
- **Personalized (history).** It employs a user history encoder based on Transformer to model the user's historical information (as shown in Fig. 3 (b)). Specifically, we use CLIP [26] to map the video clip and user posted text pair to the same space and concatenate them together as the input sequence to the user preference encoder. Being the same as our PVCG, the encoded user preference feature is used to select video clip and generate comments.
- **PVCG (cascaded).** This is a cascaded version of our proposed PVCG model. The personalized clip selection module and text generation module are optimized separately.

*5.1.2 Evaluation of Comment Generation.* To evaluate how well the generated comment is, we employ both reference-based metrics and ranking-based metrics to evaluate the generation performance. For reference-based metrics, we use BLEU [23], Meteor [6], CIDEr [29], and ROUGE [17] to evaluate the quality of generated text. For ranking-based metrics, we discriminate a good model according to its ability to rank the correct comments on the top of candidates. We construct a list of candidate comments for each user-clip pair in the test set. We pool all comments in the training set and sample negative comments in four ways:

- **Personalized comments:** The twenty comments from the same user in the training set, which can be considered as text with the same personal style but not necessarily relevant to the clip.
- **Plausible comments:** The twenty most similar comments in the training set to the video titles based on the cosine similarity of their TF-IDF vectors.
- **Popular comments:** the twenty comments randomly selected from the training set with the highest frequency.
- **Random comments:** the forty comments randomly selected from the training set.

Following VideoIC [31], we measure the ranking results with three types of metrics: (1) Recall@k: The proportion of ground truth

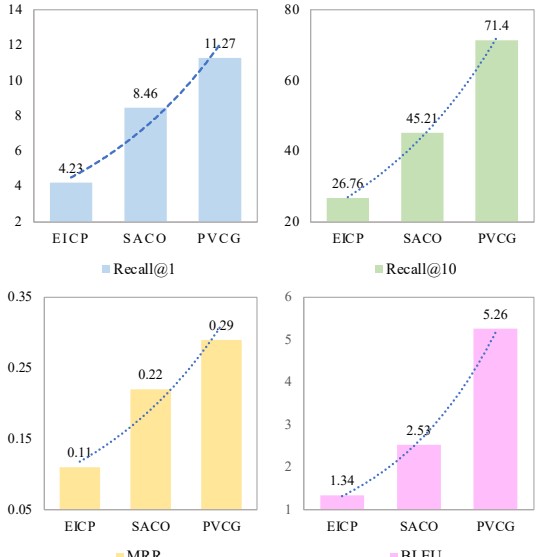

**Figure 4: Comparing the quality of generated text generated by different methods given the golden clips.**

comments in the top-k sorted text. (2) Mean Rank (MR): The mean rank of the ground truth text. (3) Mean Reciprocal Rank (MRR): The mean reciprocal rank of the ground truth text.

We compare our model with baseline methods described in Section 5.1.1 and two personalized image caption baselines.

- **EICP [28].** EICP first proposes personalized image caption task. It combines one-hot encoding personality information with standard image caption model by adding it with the text decoder input at each step. To ensure fairness, we use a pre-trained BLIP as the standard image caption model here.
- **SACO [41].** SACO is one of the SOTA models on personalized image caption. It leverages a style-aware contrastive learning to integrate style and visual information into the caption.

As EICP and SACO cannot select personalized clip, we input user actually commented clips (golden clips).

*5.1.3 Model Configuration.* The vision encoder is initialized from $ViT-B/16$ pre-trained on ImageNet [5], and the user preference encoder and text generator are initialized from $BERT_{base}$. The preference encoder and text generator share all parameters except self-attention layer and cross-attention layer. We load parameters from vision language pre-trained model – BLIP [16], and then fine-tune the model on Personalized VideoIC dataset for 20 epochs using a batch size of 256. We use AdamW optimizer with a weight decay of 0.05, and a cosine learning rate schedule. Please note that we pre-train model with contrastive loss before training to reduce the training difficulty. During pre-training, we use contrastive learning by encouraging positive clip-preference pairs to have similar representations in contrast to the negative pairs. We regard user preference features and clip features which the user is interested in as the positive pair, while the others are negative pairs. We sample negative clips across the entire dataset, rather than just sampling within the same video to avoid false negatives.

**Table 3: Comparing the quality of generated text generated by different methods. † denotes significant improvements over the PVCG (cascaded) results with $p$-value< 0.05 in t-test.**

| Settings | Ranking-based metrics | | | | | Reference-based metrics | | | |
|---|---|---|---|---|---|---|---|---|---|
| | Recall@1 ↑ | Recall@5 ↑ | Recall@10 ↑ | MR ↓ | MRR ↑ | BLEU ↑ | METEOR ↑ | ROUGE_L ↑ | CIDEr ↑ |
| MostPopular | 1.57 | 5.63 | 8.45 | 40.51 | 0.06 | 2.03 | 2.23 | 3.16 | 2.03 |
| Pers (history) | 3.63 | 6.70 | 14.50 | 34.78 | 0.09 | 3.35 | 2.33 | 2.77 | 3.39 |
| PVCG (cascaded) | 4.23 | 18.31 | 30.99 | 28.00 | 0.13 | 3.30 | 2.66 | 3.87 | 4.27 |
| PVCG(r=0) | 8.45 | **49.30** | 70.42 | 11.03 | 0.27 | 4.54 | 3.29 | 4.77 | **4.93** |
| PVCG(r=4) | **9.85**† | 47.89† | **71.83**† | **10.87** | **0.28** | **4.67**† | **3.34**† | **4.84**† | 3.51 |

**Table 4: Ablation studies by measuring both clip selection and comment generation. Here, scheduled sampling refers to confidence-aware scheduled sampling described in Section 4.1.**

| Methods | Clip selection | | | Comment Generation | | | |
|---|---|---|---|---|---|---|---|
| | Recall@1 ↑ | Recall@5 ↑ | Recall@10 ↑ | BLEU ↑ | METEOR ↑ | ROUGE_L↑ | CIDEr ↑ |
| w/o pre-train | 4.64 | 29.01 | 63.02 | 3.99 | 3.64 | 3.73 | 2.85 |
| w/o scheduled sampling | 8.45 | 29.58 | 36.62 | 4.18 | 3.10 | 4.77 | 4.81 |
| PVCG ($r = 0$) | 25.35 | 40.85 | 60.56 | 4.54 | 3.29 | 4.77 | 4.93 |
| PVCG ($r = 2$) | 25.99 | 41.52 | 63.56 | 4.56 | 3.32 | 4.80 | 4.90 |
| PVCG ($r = 4$) | **27.54** | **42.18** | **65.92** | **4.67** | **3.34** | **4.82** | 4.87 |

## 5.2 Performance on Clip Selection (RQ1)

The selection results of different models are shown in Tab. 2. First, our proposed PVCG achieves the best performance on both Recall@1 and Recall@10. It verifies the effectiveness of user preference modeling method in PVCG. The improvement over PVCG (cascaded) indicates that joint learning can improve the clip selection accuracy significantly and mutual benefit between clip selection and text generation. Second, as expected, all personalized models perform much better than MostPopular baseline which does not consider personal preference in clip selection. Third, BPR-MF method performs not as good as other personalized models, it demonstrates that using users' interaction information is not enough in video clip selection task. Forth, when compared to personalized (user id), PVCG demonstrates a higher selection accuracy, highlighting that explicit modeling user identity performs much better than introducing user identity into plain text. Fifth, PVCG outperforms Personalized (history) in terms of Recall@1 and Recall@10. This could be due to the presence of noise in the user history information, which leads to a decrease in selection accuracy.

## 5.3 Performance on Comment Generation Only (RQ2)

To evaluate the comments generation performance in multi-modal scenarios, first we evaluate the text generation quality given the user actually commented clips. As shown in Tab. 4, we compare PVCG with two personalized image caption models – EICP [28] and SACO [41]. Compared with EICP, which represents personal preference by user identity only, SACO and PVCG both perform much better in ranking-based metrics and reference-based metrics. It proves that the importance of modeling user preference considering textual information. Besides, compared with SACO, our proposed PVCG model performs better considering both ranking-based metrics and reference-based metrics. Especially, in Recall@10

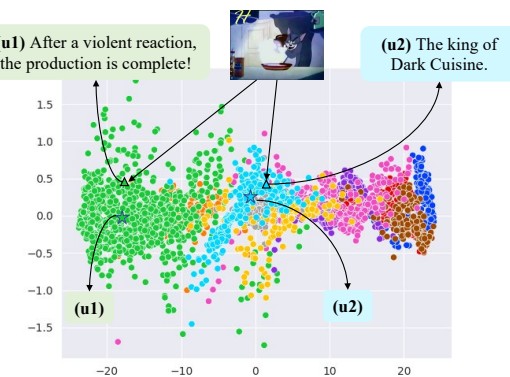

**Figure 5: Visualization of the preference representation from different users with t-SNE.**

and BLEU score, we find that PVCG brings significant improvements over baseline methods. it shows the effectiveness of our proposed training strategies and and architecture, especially in complicated multi-modal scenario.

## 5.4 Performance of Clip Selection and Comment Generation (RQ3)

We verify the performance without golden clips. As shown in Tab. 3, we have some observations. First, all personalized video to text generation methods work much better than the MostPopular baseline without personalization, which indicates that user generated text does have obvious individual preferences, thus understanding human preference is definitely helpful. Second, our proposed methods based on joint optimization of clip selection and comment generation are significantly better than the cascaded baseline. This is consistent with our observations on the Tab 2. Third, our full version PVCG method (i.e. PVCG (r=4)) performs better than the version without iterative inference in most metrics. This shows that our proposed iterative comment inference is effective.

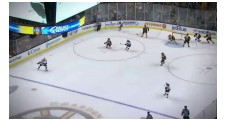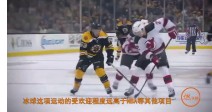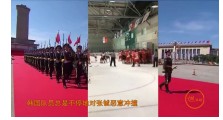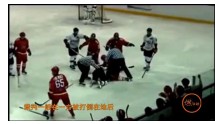

(u1) The defense line is too long, and the defense line is too long.

(u2) the ice hockey grand slam is also very painful.
(u3) It really makes sense.

(u4) Are we the champion?

(u5) Stepping on the feet 2333333

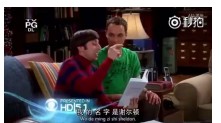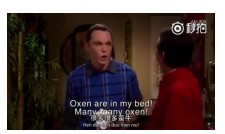

(u6) I feel his singing skills are very good.

(u5) Both people are right 233333.
(u7) I don't know what they are singing.

(u8) This is a good standard.
(u7 forced) I don't understand too much, I'm still laughing, I'm still laughing.

(u6 forced) It's so cute, it's really cute.
(u7 forced) I'm going to laugh.

**Figure 6: Generated results for different users on the test data of Personalized VideoIC. For each video, PVCG can select clip and generate comments based on user preference with PVCG in an end-to-end manner. PVCG is also flexible to generate personalized comment by giving a specific video clip as input.**

## 5.5 Ablation Study

We conduct ablation studies to investigate the impact of training strategy, validate the effectiveness of confidence-aware scheduled sampling and iterative comment generation strategy. Results are shown in Tab. 4. First, removing the pre-training stage leads to a decrease in selection accuracy and thus text generation quality. We attribute this to the fact that clips from the same video have higher similarity, making it difficult for the model to select the correct clip from those clips. The pre-training stage samples negative examples from the entire dataset, which makes it easier to select the ground truth clip. This method actually optimizes the model in an easy to hard manner, which smooths the training process, and facilitates better selection and generation performance. Second, removing confidence-aware scheduled sampling significantly lowers both the selection accuracy and the quality of the generated text. It also proves the importance of bridging the gap between the training and the inference stage in the personalized video to text generation task. Third, as the number of iterative rounds $r$ increases, both the quality of the generated text and the selection accuracy improve. In the end-to-end framework, the higher the clip selection accuracy, the more accurate the visual information utilized during comment generation, resulting in comment content and style that are more similar to the target user. Then, using the generated text as additional input of user preference encoder can better model target user preference, in turn improving the clip selection accuracy.

## 5.6 Qualitative Analysis

To better illustrate how the proposed model captures user preference and behaves in clip selection and comment generation, we do some visualization and show some cases.

Fig. 5 visualizes the user preference features from ten different users who comments on video "Dubbing of Tom and Jerry". Different colors represent preference features of different users. We observe that different users can be effectively distinguished by our user preference encoder. We can see that the preference features from the same user are very closely clustered together, regardless of whether combining corresponding comments or not. Moreover, there is a large difference in preference between user $u1$ and user $u2$. Even for comments on the same video clip, their personal preference features differ greatly.

Fig. 6 shows the selected videos and generated comments for different users. The same color refers to the text generated given same user identity. Obviously, different video clips interest different users, resulting in diverse personalized comments. We also find that the generated texts for different videos given the same user identity have similar literal preference. For example, $u5$ prefers to use "2333..." (several 3's following a 2, which means "happiness" online). Furthermore, we force the model to generate text for users on clips they are not interested in. We observe that PVCG can generate personalized text related to the clip's visual information and the user's preference. Our framework is flexible and effective to select clips, or generate text for a given clip, or do both.

## 6 CONCLUSION

In this work, we highlight the importance of personalization in video to text generation for multimodal chatbots and define personalized video to text generation task formally. By taking video comments generation as a special application, we design a personalized video to text generation model PVCG to address the task in an end-to-end manner. PVCG consists a clip selector and a comment generator, with well-designed strategies to make this framework fully differentiable. We also construct Personalized VideoIC, a new dataset of personalized video comments generation, to support the research. Experimental results demonstrate the superiority of PVCG over all baselines and the positive contributions of our proposed ideas. For future work, we plan to explore personalized clip selection considering explicit temporal modeling. How to expand the model to a real-time streaming scenario is also interesting.

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
