# OpenReview forum: "Understanding Human Preferences: Towards More Personalized Video to Text Generation"
_ACM.org/TheWebConf/2024/Conference — TheWebConf24_

### Official Review · Reviewer_LvEb · 2023-11-04

**Novelty:** 4
**Technical Quality:** 5

**Review:**

The authors discuss the importance of personalization in video to text generation models. While existing models focus on understanding video content in a general sense, they fail to capture personalized preferences, which are crucial for engaging multimodal chatbots. The document introduces the task of personalized video commenting and proposes an end-to-end framework to solve this task. The framework consists of a clip selector, responsible for predicting the clips that the user may comment on, and a text generator, which produces the comment based on the predicted clips and the user's preference. The two components are optimized in an end-to-end manner, and confidence-aware scheduled sampling and iterative inference strategies are designed to address the absence of ground truth clips in the inference phase. The document also presents a new dataset for studying personalized video comments generation and conducts extensive experiments to demonstrate the effectiveness of the proposed model.

Strengths:
1. Existing video to text models lack personalization in understanding human preferences. And the document introduces the task of personalized video commenting.
2. A framework is proposed, consisting of a clip selector and a text generator. The clip selector predicts the clips that the user may comment on, while the text generator produces the comment based on the predicted clips and the user's preference. The two components are optimized in an end-to-end manner. Confidence-aware scheduled sampling and iterative inference strategies are designed to address the absence of ground truth clips in the inference phase.
3. A new dataset is released for studying personalized video comments generation.
4. Extensive experiments are conducted to demonstrate the effectiveness of the proposed model.

Weakness:
1. The authors do not compare the proposed model with the recent video-based baseline like: video-chatgpt.
2. I am not sure if the authors are authorized to collect data from this website https://www.bilibili.com.
3. The division information of the proposed dataset for the experiments is unclear.

**Questions:**

1. Why the authors do not consider the video-based models as the baselines?
2. Are the authors authorized to collect data from this website https://www.bilibili.com?
3. Could you please provide the division information of the proposed dataset for the experiments?

**Reviewer Confidence:**

3: The reviewer is confident but not certain that the evaluation is correct

**Scope:**

4: The work is relevant to the Web and to the track, and is of broad interest to the community

---

### Official Review · Reviewer_RCq6 · 2023-11-22

**Novelty:** 5
**Technical Quality:** 3

**Review:**

This work proposed a new task called Personalized video comment, and illustrate the difficulty of this task. It also proposed a PVCG framework to solve this problem using a clip selector and a comment generator with special strategies.


Pros:

1. The author proposes a new task involving visual clip selection and text generation, and provides the construction of corresponding data sets
2. It is novel to consider using user history to select fragments as a modeling method of user interests
3. The whole content is relatively complete

Cons:

1. The task is interesting, but methodologically, especially for the visual side of the modeling description is very unclear.
2. The work does not delve into the discussion of the long-tail phenomenon in user modeling.
3. In the experiment, additional text generation methods based on video content can be incorporated as baselines to augment the persuasiveness of the article. For instance, [1].
4. The article exhibits significant issues in this paper. Further revision of the text is imperative to ensure alignment with the requisite standards for an academic paper.

[1] Ma S, Cui L, Dai D, et al. LiveBot: Generating Live Video Comments Based on Visual and Textual Contexts[J]. 2019.

While the authors introduce a new task along with a simple and effective method to address the associated challenges, the experiments and baselines appear somewhat limited. This restricts the ability to convincingly demonstrate the effectiveness of their work. Additionally, there are issues in the writing. Therefore, a weak rejection is recommended.

**Questions:**

1. The author describes the ENC module based on vision transformer in section 4.3, but does not describe in detail how to use an image encoder to encode video.
2. The introduction section of the article lacks coherence.
3. A long scenario description appears in the first paragraph of the problem definition.

**Reviewer Confidence:**

3: The reviewer is confident but not certain that the evaluation is correct

**Scope:**

3: The work is somewhat relevant to the Web and to the track, and is of narrow interest to a sub-community

---

### Official Review · Reviewer_ZKoX · 2023-11-23

**Novelty:** 6
**Technical Quality:** 6

**Review:**

In this paper, the authors present a framework for personalized video-to-text generation. The framework includes two components, namely the clip selector and text generator, which are jointly trained. To address the clip absence problem, they introduce a confidence-aware sampling and iterative inference strategy. Additionally, a new dataset is introduced to support this novel scenario.

Advantages:

- The concept of personalization in video commenting is innovative. The framework is well-crafted to address this challenge, involving multimodal comprehension and generation. It stands as a successful case for personalized generation in the recommendation domain.
- The paper is easily understandable, and the illustrations are clear.
- The framework undergoes thorough examination and testing on the new dataset, incorporating ablation studies and visualizations.


Disadvantages:

- Implementation details are lacking, including information about the training machine and feature dimensions.
- An analysis of time and space requirements is necessary.
- Further investigation and clarification are needed regarding the inferiority of the Recall@5 metric compared to the Pers (history) model.

**Questions:**

Q1. Can you offer additional implementation details? Will the data and code be made available after the paper is accepted? Reproducibility is a significant concern for me.

Q2. Could you incorporate an analysis of time and space complexity?

Q3. Can you provide an explanation for why the Pers (history) model outperforms PVCG on the Recall@5 metric, as indicated in Table 2?

**Ethics Review Description:**

/

**Reviewer Confidence:**

2: The reviewer is willing to defend the evaluation, but it is likely that the reviewer did not understand parts of the paper

**Scope:**

4: The work is relevant to the Web and to the track, and is of broad interest to the community

---

### Official Review · Reviewer_BXrP · 2023-11-23

**Novelty:** 4
**Technical Quality:** 5

**Review:**

The paper proposed a new video-to-text framework for streaming video commenting with user personalizations. The framework consists of a user preference modeling module and a comment generation module. Upon a new annotated video dataset, the proposed framework mostly acheived the best performance.

**Questions:**

1. Can authors provide some real-world example appliations that provide real-time streaming videos, allow user comments but no comment visibility from other users? I think most real-time streaming services have a user comment pool that is either in the side of the video or even directly on the videos. Then the CF could be possible.
2. For Section 4.3, what is the difference between the proposed method and the regular item-level recommendation if we consider each video clip as an item? Also, can authors further explain the need of the CE loss when the top-1 user-clip representation can realdy do so?
3. For experiments, I wonder if there is any trade off between the personalization and diversity. Say, is there any case where the trained model will just generate comments that are exactly the same as the training comments of a user to achieve the high persoanlization?

**Reviewer Confidence:**

3: The reviewer is confident but not certain that the evaluation is correct

**Scope:**

2: The connection to the Web is incidental, e.g., use of Web data or API

---

### Decision · Program_Chairs · 2024-01-22

**Decision:**

Accept

**Comment:**

Scores are a little mixed but ultimately lean positive. Reviewers raise issues about comparisons, details, additional evidence, etc., though these seem not to be dealbreakers, and given the overall positive scores there seems to be more evidence to recommend acceptance.